# Association between Self-Reported Exposure to Alcohol Advertisements and Drinking Behaviors: An Analysis of a Population-Based Survey in Thailand

**DOI:** 10.3390/ijerph182111271

**Published:** 2021-10-27

**Authors:** Phagapun Boontem, Udomsak Saengow

**Affiliations:** 1Faculty of Nursing, Srinakharinwirot University, Nakhon Nayok 26120, Thailand; phagapun@g.swu.ac.th; 2Research Institute for Health Sciences, Walailak University, Nakhon Si Thammarat 80160, Thailand; 3Center of Excellence in Data Science for Health Study, Walailak University, Nakhon Si Thammarat 80160, Thailand; 4School of Medicine, Walailak University, Nakhon Si Thammarat 80160, Thailand

**Keywords:** alcohol advertisement, alcohol advertising exposure, alcohol consumption, heavy drinking, association, cross-sectional study

## Abstract

The relationship between alcohol advertising and drinking has been demonstrated in many studies. Most studies were conducted on adolescents or young adults. Thailand has strict regulations on alcohol advertisements. This study aimed to examine associations between exposure to alcohol advertisements and drinking behaviors, i.e., past-year drinking and past-year heavy drinking, using data from a population-based survey in Thailand. The survey participants were Thai citizens aged 15 or older. Logistic regression was used to investigate the associations. The primary explanatory variable was self-reported exposure to alcohol advertisements. Covariates in the regression models included sex, age, and education. Self-reported exposure to alcohol advertisements was associated with past-year drinking (OR, 1.16; 95% CI, 1.07–1.27), past-year heavy drinking (OR, 1.35; 95% CI, 1.28–1.41), and past-year heavy drinking among drinkers (OR, 1.51; 95% CI, 1.43–1.60). Male sex, working age, and secondary education or a diploma were associated with higher odds of past-year drinking and past-year heavy drinking. In this study, self-reported exposure to alcohol advertisements was shown to be associated with past-year drinking and past-year heavy drinking among a population aged 15 years or older.

## 1. Introduction

Alcohol consumption is attributed to more than 200 diseases and injuries. Health consequences related to alcohol use include maternal health, child development, cancer, cardiovascular diseases, injuries, violence, mental health, TB, and HIV/AIDS, among others. In 2016, there were 3 million alcohol-related deaths worldwide. In the same year, alcohol was responsible for 132.6 million disability-adjusted life years (DALYs). Premature death accounted for more than 80% of the DALYs associated with alcohol [1].

Studies from several countries found a positive association between exposure to alcohol advertisements and alcohol consumption. A survey of 909 adolescents from Chile reported a 41% increase in the likelihood of being drinkers and an 85% increase in the risk of being problematic drinkers in adolescents with recent exposure to alcohol advertisements [2]. The daily or almost daily exposure to alcohol advertisements was associated with current alcohol use (odds ratio (OR), 1.6; 95% confidence interval (CI), 1.0–2.5) and drunkenness (OR, 2.3; 95% CI, 1.4–3.8) according to a cross-sectional study of 3806 school children from Cambodia [3]. A household survey of 955 South Africans aged 18–65 years old found that alcohol advertising using SMS and free alcohol offers was associated with heavy drinking [4]. A cross-national survey of 9038 school children from Germany, Italy, the Netherlands, and Poland demonstrated that exposure to online alcohol advertising and owning alcohol-branded items increased the likelihood of ever use of alcohol and past-month binge drinking [5]. An online survey of 3399 UK adolescents found that a higher level of past-month exposure to alcohol marketing was associated with a higher-risk drinking pattern. Owning alcohol-branded items was associated with a higher-risk drinking pattern among drinkers and a higher susceptibility to drink among never drinkers [6]. A recent large study of 54,671 US adults aged 21 years and older found that a higher volume of alcohol advertising exposure was associated with past-month consumption of alcohol and, among past-month drinkers, more drinks consumed [7]. In addition to the evidence from these cross-sectional studies, a systematic review of 12 cohort studies regarding alcohol consumption in youths found that the level of alcohol marketing exposure was positively associated with alcohol use and a higher-risk drinking pattern among youths [8]. Regarding the rise in digital marketing, a recent systematic review found that engagement in digital alcohol marketing activities was associated with alcohol consumption [9]. Hence, a relationship between alcohol advertising and drinking was demonstrated in studies from various countries. The majority of studies focused on adolescents or young adults.

In Thailand, regulation of alcohol advertising and marketing is included as a measure under the altering social norms toward alcohol and reducing drinking motivation strategy, one of the five strategies of the 2010 National Alcohol Strategy [10]. The primary law regulating alcohol advertising and marketing is the Alcoholic Beverage Control Act, B.E. 2551 (2008). Following the enactment of the Alcoholic Beverage Control Act, B.E. 2551 (2008), most types of alcohol advertising in Thailand have been banned. Section 32 of the act [11], which is a regulation of alcohol advertising, states 


*“No person shall advertise or display, directly or indirectly, the name or trademark of any alcoholic beverage in a manner showing the properties thereof or inducing another person to drink.*



*Advertisements or public relations provided by the manufacturer of any kind of alcoholic beverage shall only be made for giving information thereof or giving social creative knowledge without displaying any illustration of such alcoholic beverage or its package, except for the display of a symbol of such alcoholic beverage or that of its manufacturer as prescribed by the Ministerial Regulation.*



*The provisions of paragraph one and paragraph two shall not apply to any advertisement broadcast from outside of the Kingdom.”*


The act effectively prohibits all direct advertising of alcoholic beverages. It only allows for a brief presentation of the logo with messages unrelated to drinking. Recently, surrogate advertising has been used by the alcohol industry in Thailand. Soft drink logos (such as drinking water, soda, and flavored soft drinks) were recently changed to resemble the logos of alcoholic beverages owned by the same conglomerate [12]. These soft drinks are promoted in place of alcoholic beverages through all media channels, including television, newspapers, billboards, and social media. In addition to the regulation, there is an annual social movement campaign encouraging three months of sobriety, in which millions of drinkers participate [13]. This campaign aims to counteract the effect of advertising and marketing on altering norms toward drinking. It is included in the 2010 National Alcohol Strategy under the same strategy as the regulation of alcohol advertising and marketing [10].

In the context of the Thai alcohol policy regarding alcohol advertising described above, this study aimed to examine the association between exposure to alcohol advertisements and drinking behaviors (i.e., past-year drinking and past-year heavy drinking) using data from a population-based survey of Thai citizens aged 15 years or older.

## 2. Materials and Methods

### 2.1. Study Design

This study was an analysis of data from the 2017 Smoking and Drinking Behavior Survey (SADBeS17), which was conducted by the National Statistical Office, Thailand. Data from part III of the survey, which was related to alcohol consumption and its consequences, were used in the analysis. The protocol of this study was approved by the Human Research Ethics Committee of Walailak University, Nakhon Si Thammarat, Thailand (WU-EC-MD-3-473-63).

### 2.2. Data Source

The SADBeS17 is a population-based survey of Thai citizens aged 15 or older. The stratified two-stage sampling technique was used. Each province was considered a stratum. There were a total of 77 strata. In each stratum, 2315 enumeration areas were chosen at random from a total of 129,440 enumeration areas. Households were randomly selected from the enumeration areas. 

The eligibility criteria included being 15 years old or older, having Thai citizenship, and being able to communicate fluently in Thai. All eligible individuals in selected households were invited to participate in the survey. Informed consent was obtained from all survey participants. For participants aged less than 18 years, consent was given by their guardians and themselves. The total number of survey participants was 92,015. The response rate was 93.5%. There were 89,154 participants with complete data. The analysis included only records with complete data. The data were collected between May and July 2017.

The survey items included demographic characteristics (age, sex, nationality, education level, marital status, occupational, etc.), smoking-related items, and alcohol-related items. The outcome variables in this study were drinking and past-year heavy drinking. Self-reported exposure to alcohol advertisements was the main explanatory variable. Covariates included sex, age, and education level. These covariates were selected as they are recognized determinants of alcohol consumption [1]. 

### 2.3. Data Management

The variables used in this analysis included drinking status, past-year heavy drinking, self-reported exposure to alcohol advertisements, sex, age group, and education level. 

The primary outcome was past-year drinking. The item for past-year drinking was “Had you ever drunk alcoholic beverages in the past 12 months?” The responses were classified into three levels: no, occasionally (once a month or less frequently), and regularly (at least once a week). Participants who drank occasionally or regularly were classified as past-year drinkers based on their responses to this item. The secondary outcome was past-year heavy drinking. The item for past-year heavy drinking was “How often had you drunk heavily (5 drinks or more) in a short period of time in the past 12 months?” The responses were classified into three levels: no, occasionally (once a month or less frequently), and regularly (at least once a week). Both drinking variables were dichotomized in order to be used as dependent variables in the binary logistic regression. The responses “occasionally” and “regularly” were categorized as “yes”, while the response “no” remained unchanged.

Self-reported exposure to alcohol advertisements was the main explanatory variable. The item was “Had you been exposed to alcohol advertisements in the last 30 days?” There were three possible responses to this question: yes, no, and uncertain. There was no further explanation provided regarding alcohol advertising. Hence, participants considered what they perceived to be alcohol advertisements by themselves. The sex variable had two levels: male and female. Age was classified into five categories: 15–19, 20–30, 31–45, 46–60, and 61+. There were four levels of education: primary education or lower level, secondary education or diploma, bachelor’s degree or higher, and missing.

### 2.4. Statistical Analysis

Survey participants were described using descriptive statistics (mean, standard deviation, and percentage). The percentages of past-year drinkers, past-year heavy drinkers (among the total population), and past-year heavy drinkers among drinkers were computed. Binary logistic regression was performed to examine the relationship between self-reported exposure to alcohol advertisements and both drinking variables (i.e., past-year drinking and past-year heavy drinking). 

The regression model for past-year drinking had self-reported exposure to alcohol advertisements, sex, age, and education level as covariates (Model 1). Data from 89,154 participants with complete information were used in this model. For past-year heavy drinking, two regression models were performed (Model 2 and Model 3). Both models had the same set of covariates, including self-reported exposure to alcohol advertisements, sex, age, and education level. Model 2 examined the association between self-reported exposure to alcohol advertisements and past-year heavy drinking among the total participants. As in Model 1, data from all participants with complete information were used. Model 3 examined an association between self-reported exposure to alcohol advertisements and past-year heavy drinking among past-year drinkers. Data from 23,073 past-year drinkers were used in Model 3. Odds ratios (ORs) and 95% CIs were estimated from the regression models to indicate the magnitude of associations. The level of significance was set at 5%. All analyses were unweighted. The R statistical language version 4.0.3 was used to perform the statistical analysis.

## 3. Results

### 3.1. Characteristics of Participants

Over half of the participants were women. The average age was 47.5 years old. More than half attained primary education or lower, whereas 11.3% attained a bachelor’s degree or higher. One third had been exposed to alcohol advertisements; 3.8% could not recall whether they had been exposed or not. One fourth were past-year drinkers; 11.2% drank regularly in the past year. About 10% of the participants were past-year heavy drinkers; more than half of the heavy drinkers regularly engaged in heavy drinking episodes (Table 1).

### 3.2. Past-Year Drinkers and Past-Year Heavy Drinkers

The proportion of past-year drinkers in men was considerably higher than in women at 45.1% versus 9.7%. The 31–45 age group had a higher proportion of past-year drinkers; the proportion was the lowest in the 15–19 age group. Secondary education or a diploma was the education level with the highest proportion of past-year drinkers. Participants that had no exposure to alcohol advertisements represented a lower proportion of past-year drinkers than those who had exposure. A similar pattern was observed for past-year heavy drinking. Participants with male sex, working age, and secondary education or a diploma had a higher proportion of past-year heavy drinking compared to the other groups. A higher proportion of past-year heavy drinkers was observed in participants with exposure to alcohol advertisements than those without exposure, with the highest proportion represented by those who were uncertain about their exposure (Table 2).

### 3.3. Factors Associated with Past-Year Drinking 

Table 3 displays the results of the regression model for past-year drinking (Model 1). Self-reported exposure to alcohol advertisements increased the odds of drinking by 16%. Men were eight times more likely than women to be past-year drinkers. Participants in the working age groups were three to four times more likely to drink in the past year than those in the 61+ age group. Participants with the highest education level had slightly lower odds of drinking in the past year than those with primary education or lower, while those with secondary education or a diploma had higher odds of drinking.

### 3.4. Factors Associated with Past-Year Heavy Drinking

Table 4 presents the results of two regression models regarding the association between self-reported exposure to alcohol advertisements and past-year heavy drinking among the total participants (Model 2) and among past-year drinkers (Model 3). In Model 2, self-reported exposure to alcohol advertisements was associated with a 35% increase in the odds of past-year heavy drinking, whereas participants who were uncertain about their exposure had a 68% increase in the odds when compared to those who were not exposed. Men were 12 times more likely than women to be past-year heavy drinkers. The 31–45 age group had the highest odds of past-year heavy drinking. When compared to those with primary education or lower, those with the highest education level had lower odds of past-year heavy drinking, while those with secondary education or a diploma had higher odds of past-year heavy drinking. 

In Model 3, self-reported exposure to alcohol advertisements was associated with a 51% increase in the odds of past-year heavy drinking. Past-year drinkers in all other age groups had higher odds of past-year heavy drinking than those aged 61+. Drinkers with a bachelor’s degree or higher had lower odds of past-year heavy drinking than those in the lowest education group.

## 4. Discussion

In this study, the association between alcohol advertising exposure and drinking behaviors was investigated using data from a population-based survey. We demonstrate that self-reported exposure to alcohol advertisements was associated with past-year drinking and past-year heavy drinking adjusted for sex, age, and education. Male sex, working age, and secondary education or a diploma were associated with greater odds of past-year drinking and past-year heavy drinking.

The current study found that self-reported exposure to alcohol advertisements was associated with past-year drinking (OR, 1.16; 95% CI, 1.07–1.27), past-year heavy drinking (OR, 1.35; 95% CI, 1.28–1.41), and past-year heavy drinking among drinkers (OR, 1.51; 95% CI, 1.43–1.60) in participants aged 15 or older. Similarly, a previous cross-sectional survey of 1200 university students in Bangkok, Thailand, reported a positive association between self-reported alcohol advertising exposure and the frequency of past-year drinking [14]. Self-reported exposure to alcohol advertising increased the likelihood of ever drinking (relative risk (RR), 1.41; 95% CI, 1.10–1.80) and being problematic drinkers (RR, 1.85; 95% CI, 1.40–2.44) in a study of 1076 Chilean adolescents [2]. An analysis of a cross-sectional survey of 3806 school children in Cambodia found that almost daily/daily exposure to alcohol advertising (self-reported) was associated with alcohol use in the last 30 days (OR, 1.61; 95% CI, 1.03–2.51) and ever having drunkenness (OR, 2.30; 95% CI, 1.40–3.77) [3]. An online cross-sectional survey of 3999 UK adolescents reported that a medium to high level of self-reported alcohol marketing exposure (referred to as “alcohol marketing awareness” in the original article) was associated with high-risk drinking (i.e., AUDIT-C score ≥ 5; OR, 1.43–2.18) among past-year drinkers, but not with susceptibility to drink among teetotalers [6]. Summarily, self-reported exposure to alcohol advertisements was positively associated with various measures of drinking behaviors, with the exception of teetotalers’ susceptibility to drink. 

Whereas the studies discussed in the preceding paragraph used self-reported exposure to any types of alcohol advertising as the exposure variable in multivariate analysis, there were studies that examined the effects of exposure to specific types of alcohol advertising or marketing. A cross-sectional study of 955 participants from a municipality in South Africa reported that SMS advertising and free alcohol offers were associated with a higher chance of heavy drinking in the past six months. Large posters/billboards, sport and music sponsorship, signs or posters, TV, radio, famous people, magazines/newspapers, and email advertising, on the other hand, were not associated with heavy drinking [4]. A cross-sectional study of 4413 adolescents in Victoria, Australia, found that self-reported exposure (awareness) to alcohol advertising via billboards/newspapers/magazines and alcohol merchandise ownership increased the chance of past-month alcohol use and past-week heavy drinking. Past-month alcohol use and past-week heavy drinking were not associated with advertising on television, at sporting events, on the internet, or through sports sponsorship [15]. An online cross-sectional survey of 9038 school students from four European countries reported that self-reported online alcohol advertising exposure and ownership of alcohol-branded items were associated with an increased chance of ever drinking and past-month heavy drinking. This study also found a dose–response relationship between overall exposure to alcohol advertisements and ever drinking and past-month heavy drinking [5]. A cross-sectional study of 2257 school students in Zambia found that being given free alcohol was associated with ever having drunkenness and ever having drinking-related problems. The study found no association between other types of alcohol marketing (i.e., using actors and billboards) and either drinking outcome [16]. These findings suggest that different types of advertising or marketing activities may have varying effects on drinking behaviors. This could be due to the differences in the context of each study. It could also be because of a qualitative aspect of the advertisement, such as its design, provision, and timing. Nonetheless, free alcohol offers and ownership of alcohol merchandise were consistently associated with drinking behaviors across studies (where these types of advertising were included as exposure variables).

The proportion of participants in this study who were exposed to alcohol advertisements was 32.7%, which was lower than the proportions reported in other studies. The percentage of past-month exposure to alcohol advertising in Cambodian school children was 81.4% [3]. More than 40% of UK adolescents were exposed to alcohol advertising on television in the past week [6]. Almost half of the participants in a survey of South Africans reported that they were frequently exposed to alcohol advertisements [4]. This was likely due to the effect of Section 32 of Thailand’s Alcoholic Beverage Control Act, B.E. 2551 (2008), which prohibits most forms of alcohol advertising.

Our findings are in agreement with those of other studies, which found that exposure to alcohol advertisements was generally associated with a higher chance of drinking and heavy drinking. Due to a lack of information on each type of alcohol advertising exposure in the SADBeS17, we were unable to explore the effect of each type of advertising, which has been shown to have a different effect on drinking behaviors. Most studies on this topic were conducted using data from adolescents or young adults, whereas the data used in this study were collected from the general population aged 15 years or older. As a result, the findings of this study can be generalized to a wider population compared to those from previous studies.

Our finding that men were more likely to be past-year drinkers and past-year heavy drinkers fits with the general pattern of drinking behaviors [1]. The findings on drinking and heavy drinking by age group are also consistent with drinking patterns in middle-income Asian countries, where adolescents and the elderly had less likelihood of being past-year drinkers and heavy drinkers than working-age populations [1,17]. In general, socioeconomic status is associated with greater alcohol consumption [1]. The relationship between education level (one of the indicators of socioeconomic status) and past-year drinking and past-year heavy drinking did not follow that pattern. Participants in the current study who had secondary education or a diploma had the highest odds of being past-year drinkers and past-year heavy drinkers. Nonetheless, this pattern was observed in a study that analyzed data from two population-based surveys conducted in Serbia and Hungary. Survey participants with secondary education had the highest chance of past-year drinking. This pattern was observed in both countries [18]. A study conducted in the United States reported a positive association between education and ever drinking. The sub-group analysis revealed that this pattern was found in Whites, but not in African American participants. African American participants with a high school diploma had a higher chance of ever drinking than those with no diploma or a college education [19]. This demonstrates that the relationship between education level and drinking varies across countries or populations.

The major strength of this study is that it used data from a population-based survey, whereas most studies exploring the relationship between alcohol advertisement and drinking behaviors used data from a narrow age range (adolescents and/or young adults). The findings of this study can be applied to a wider population. This study does have some limitations. The exposure to alcohol advertisements was determined by self-report. As a result, some participants may be unaware of their exposure, but it may still affect their drinking behaviors. This issue was partially resolved by having “uncertain” as a level of the exposure variable. Participants who were unsure about their exposure were more similar to those with exposure to alcohol advertisements than those without exposure in terms of their drinking behaviors. The survey used a 12-month duration for assessment of drinking behavior. This duration is relatively long, and it is possible that participants may change their drinking pattern during this period. The past 30-day or 4-week duration, as employed in several other studies, should be considered in future research on this topic. The survey did not collect information on exposure to each type of advertisement. As shown in a number of studies that different types of advertising and marketing had a varying magnitude of effect on alcohol consumption, data on the types and degree of advertising exposure should be collected and used as explanatory variables to enhance the understanding of the effect of alcohol advertising and marketing on alcohol consumption in Thailand. Another limitation was the cross-sectional nature of the data, which did not allow causal inference. Only the association between variables could be examined. Further studies on this topic in Thailand should employ a longitudinal design and include exposure to different types of advertisements.

## 5. Conclusions

Using data from a population-based survey in Thailand, a middle-income country with strict regulations on alcohol advertisements, this analysis shows that self-reported exposure to alcohol advertisements was associated with increasing odds of past-year drinking and past-year heavy drinking. Further studies exploring the effect of each type of advertisement using more sensitive alcohol consumption measurements are encouraged to gain more insight into the effect of alcohol advertisements on alcohol consumption.

## Figures and Tables

**Table 1 ijerph-18-11271-t001:** Characteristics of participants in the analysis.

Characteristic	Frequency	%
Sex		
Male	40,724	45.7
Female	48,430	54.3
Age (years)	Mean = 47.5 (SD = 17.5)	
15–19	6046	6.8
20–30	11,458	12.9
31–45	22,510	25.2
46–60	27,757	31.1
61+	21,383	24.0
Education level		
Primary education or lower	49,500	55.5
Secondary education or diploma	29,336	32.9
Bachelor’s degree or higher	10,093	11.3
Missing	225	0.3
Self-reported exposure to alcohol advertisements		
Exposed	29,128	32.7
Not exposed	56,637	63.5
Uncertain	3389	3.8
Past-year drinking		
No	66,081	74.2
Occasionally	13,057	14.6
Regularly	10,016	11.2
Past-year heavy drinking		
No	79,927	89.65
Occasionally	3943	4.42
Regularly	5284	5.93

SD, standard deviation.

**Table 2 ijerph-18-11271-t002:** Proportions of past-year drinkers, past-year heavy drinkers, and past-year heavy drinkers among drinkers by characteristics.

Characteristic	%		
	Past-Year Drinkers (*n* = 89,154)	Past-Year Heavy Drinkers (*n* = 89,154)	Past-Year Heavy Drinkers among Drinkers (*n* = 23,073)
Total	25.9	10.4	40.0
Sex			
Male	45.1	20.2	44.7
Female	9.7	2.1	21.5
Age (years)			
15–19	12.1	4.8	39.5
20–30	31.5	14.1	44.8
31–45	34.4	14.8	42.9
46–60	29.2	11.3	38.9
61+	13.5	4.0	29.4
Education level			
Primary education or lower	23.5	9.0	38.5
Secondary education or diploma	30.7	13.2	43.0
Bachelor’s degree or higher	23.7	8.6	36.1
Missing	16.9	8.0	47.4
Self-reported exposure to alcohol advertisements			
Exposed	27.7	12.7	45.9
Not exposed	24.8	8.9	35.9
Uncertain	28.5	14.3	50.3

**Table 3 ijerph-18-11271-t003:** Factors associated with past-year drinking (Model 1; *n* = 89,154).

Factor	OR	95% CI	*p*-Value
Self-reported exposure to alcohol advertisements			
Exposed	1.16	1.07–1.27	0.001
Uncertain	1.02	0.99–1.06	0.197
Not exposed	ref		
Sex			
Male	8.33	8.02–8.65	0.000
Female	ref		
Age (years)			
15–19	0.87	0.79–0.97	0.013
20–30	3.46	3.22–3.71	0.000
31–45	4.01	3.79–4.24	0.000
46–60	3.03	2.88–3.19	0.000
61+	ref		
Education level			
Secondary education or diploma	1.10	1.06–1.15	0.000
Bachelor’s degree or higher	0.82	0.76–0.87	0.000
Missing	0.41	0.28–0.60	0.000
Primary education or lower	ref		

**Table 4 ijerph-18-11271-t004:** Factors associated with past-year heavy drinking.

Factor	Model 2 (*n* = 89,154)			Model 3 (*n* = 23,073)		
	OR	95% CI	*p*-Value	OR	95% CI	*p*-Value
Self-reported exposure to alcohol advertisements						
Exposed	1.35	1.28–1.41	0.000	1.51	1.43–1.60	0.000
Uncertain	1.68	1.51–1.87	0.000	1.74	1.52–1.99	0.000
Not exposed	ref			ref		
Sex						
Male	12.23	11.43–13.09	0.000	3.12	2.89–3.37	0.000
Female	ref			ref		
Age (years)						
15–19	1.06	0.90–1.24	0.488	1.34	1.11–1.62	0.003
20–30	4.04	3.65–4.48	0.000	1.91	1.69–2.15	0.000
31–45	4.46	4.09–4.86	0.000	1.93	1.74–2.13	0.000
46–60	3.31	3.05–3.59	0.000	1.65	1.51–1.82	0.000
61+	ref			ref		
Education level						
Secondary education or diploma	1.06	1.00–1.12	0.042	0.96	0.90–1.02	0.178
Bachelor’s degree or higher	0.73	0.68–0.80	0.000	0.77	0.70–0.85	0.000
Missing	0.58	0.35–0.96	0.033	1.11	0.58–2.13	0.749
Primary education or lower	ref			ref		

## Data Availability

The authors obtained the data used in this study from the Center for Alcohol Studies with the permission under the contract of this study. Request for the data can be made to the corresponding author (saengow.udomsak@gmail.com). Access to the data must be permitted by the Center for Alcohol Studies.

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
