# Peer review of "Association between Self-Reported Exposure to Alcohol Advertisements and Drinking Behaviors: An Analysis of a Population-Based Survey in Thailand"

_ijerph, 2021, doi:10.3390/ijerph182111271_

Round 1

Reviewer 1 Report

The article addresses a very relevant object of analysis relating to alcohol (one of the main health risk factors) in relation to advertising and its incidence on consumption, and the main outcome is very significative.

Overall, I consider it to be an interesting work with enriching findings for the field of knowledge. However, I consider pertinent for the authors to review the following aspects:

  • The article lacks a theoretical basis and the few references are not particularly recent. It is necessary to increase the introduction by referring to the recent scientific literature. Examples: Muela-Molina, Perelló-Oliver, García-Arranz (2021):“Alcohol Advertising Regulation for the Adult Population: An Analysis of Distilled Spirits Radio Endorsements in Spain”, or Niedderdeppe et al (2021): “Estimated televised alcohol advertising exposure in the past year and associations with past 30-day drinking behavior among American adults: results from a secondary analysis of large-scale advertising and survey data”, among others.
  • While it is true that the authors provide some contextual data on alcohol consumption in the world in the Introduction, they should include more specific data to Thailand to frame better their work.
  • The characteristics presented in Table 1 should be more described in that section.
  • As a strong point,  the Discussion is very well-considered and is highly consistent in its strengths and limitations.
  • The conclusions must be expanded. They are clearly insufficient

Author Response

Point 1: The article lacks a theoretical basis and the few references are not particularly recent. It is necessary to increase the introduction by referring to the recent scientific literature. Examples: Muela-Molina, Perelló-Oliver, García-Arranz (2021):“Alcohol Advertising Regulation for the Adult Population: An Analysis of Distilled Spirits Radio Endorsements in Spain”, or Niedderdeppe et al (2021): “Estimated televised alcohol advertising exposure in the past year and associations with past 30-day drinking behavior among American adults: results from a secondary analysis of large-scale advertising and survey data”, among others.

Response 1: The Introduction was expanded to include more detail of each study cited. More recent studies were added.

Point 2: While it is true that the authors provide some contextual data on alcohol consumption in the world in the Introduction, they should include more specific data to Thailand to frame better their work.

Response 2: More background of Thai alcohol policy regarding alcohol advertising was added to the Introduction.

Point 3: The characteristics presented in Table 1 should be more described in that section.

Response 3: The text description of Table 1 was expanded.

Point 4: As a strong point, the Discussion is very well-considered and is highly consistent in its strengths and limitations.

Response 4: Thank you for this encouraging words. Some limitations were added following the comments by the other reviewer.

Point 5: The conclusions must be expanded. They are clearly insufficient

Response 5: The conclusions was expanded.

Reviewer 2 Report

Dear authors,

Congratulations on your effort. I suggest you proofread the English in your article (ex: line 13 most of studies, line 85 was be given). In addition, I have some questions/comments:

-Would have been interesting to see some comments on your discussion about your choice of variables. You putting together as past-year drinkers participants with very different drinking habits deserves, in my opinion, some considerations as to how it could have biased your results and how future research could take that into consideration was needed in your discussion. Maybe also mention a possible dose-response pattern in case the variable was analysed in a different way.

-In the discussion could also be interesting and add some value to your article, discussing how different types of advertisement (given the very broad age range of participants) could have a different effect. And if so again, how could this be addressed in future research.

-In your methodology, you could act how your covariates were selected: was there a systematic process of selection? 

Author Response

Point 1: Would have been interesting to see some comments on your discussion about your choice of variables. You putting together as past-year drinkers participants with very different drinking habits deserves, in my opinion, some considerations as to how it could have biased your results and how future research could take that into consideration was needed in your discussion. Maybe also mention a possible dose-response pattern in case the variable was analysed in a different way.

Response 1: This issue was discussed as a limitation of the study.

Point 2: In the discussion could also be interesting and add some value to your article, discussing how different types of advertisement (given the very broad age range of participants) could have a different effect. And if so again, how could this be addressed in future research.

Response 2: This issue was discussed as a limitation of the study.

Point 3: In your methodology, you could act how your covariates were selected: was there a systematic process of selection? 

Response 3: The rationale for selecting covariates was added.

Point 4: I suggest you proofread the English in your article.  

Response 4: The revision was proofread.